# Radiation Potentiates Monocyte Infiltration into Tumors by Ninjurin1 Expression in Endothelial Cells

**DOI:** 10.3390/cells9051086

**Published:** 2020-04-28

**Authors:** Ju-Hee Kang, Jong Kyu Woo, Yeong-Su Jang, Seung Hyun Oh

**Affiliations:** College of pharmacy, Gachon University, Incheon 21936, Korea; applekjh0503@hanmail.net (J.-H.K.); apoptosis@snu.ac.kr (J.K.W.); winner832000@hanmail.net (Y.-S.J.)

**Keywords:** Ninjurin1, radiotherapy, endothelial cells, monocytes, cancer, p53

## Abstract

Radiation is a widely used treatment for cancer patients, with over half the cancer patients receiving radiation therapy during their course of treatment. Considerable evidence from both preclinical and clinical studies show that tumor recurrence gets restored following radiotherapy, due to the influx of circulating cells consisting primarily of monocytes. The attachment of monocyte to endothelial cell is the first step of the extravasation process. However, the exact molecules that direct the transmigration of monocyte from the blood vessels to the tumors remain largely unknown. The nerve injury-induced protein 1 (Ninjurin1 or Ninj1) gene, which encodes a homophilic adhesion molecule and cell surface protein, was found to be upregulated in inflammatory lesions, particularly in macrophages/monocytes, neutrophils, and endothelial cells. More recently Ninj1 was reported to be regulated following p53 activation. Considering p53 has been known to be activated by radiation, we wondered whether Ninj1 could be increased in the endothelial cells by radiation and it might contribute to the recruiting of monocytes in the tumor. Here we demonstrate that radiation-mediated up-regulation of Ninj1 in endothelial cell lines such as human umbilical vein endothelial cells (HUVECs), EA.hy926, and immortalized HUVECs. Consistent with this, we found over-expressed Ninj1 in irradiated xenograft tumors, and increased monocyte infiltration into tumors. Radiation-induced Ninj1 was transcriptionally regulated by p53, as confirmed by transfection of p53 siRNA. In addition, Ninj1 over-expression in endothelial cells accelerated monocyte adhesion. Irradiation-induced endothelial cells and monocyte interaction was inhibited by knock-down of Ninj1. Furthermore, over-expressed Ninj1 stimulated MMP-2 and MMP-9 expression in monocyte cell lines, whereas the MMP-2 and MMP-9 expression were attenuated by Ninj1 knock-down in monocytes. Taken together, we provide evidence that Ninj1 is a key molecule that generates an interaction between endothelial cells and monocytes. This result suggests that radiation-mediated Ninj1 expression in endothelial cells could be involved in the post-radiotherapy recurrence mechanism.

## 1. Introduction

Radiotherapy is one of the major therapies for malignant tumors characterized by uncontrolled growth and metastasis [1,2]. Radiation treatment-mediated cell death mechanism is primarily dependent on inducing highly reactive free radicals which produces damage to genomic DNA. Although the death of tumor cells via cytotoxic effects of free radicals is an important response of tumors to radiotherapy, emerging evidence indicates that the ultimate outcome of the treatment is also critically affected by radiation. Clinical and pre-clinical evidences indicate that radiotherapy might promote a metastatic behavior of cancer cells, and the irradiated host microenvironment might exert tumor promoting effects [2,3,4,5]. Hence, a careful analysis in the alterations of the tumor environment is required, including effects on endothelial cells, leukocytes, monocytes, and fibroblasts. Conventional fractionated radiotherapy causes relatively little damage in tumor vasculatures, thereby enhancing angiogenesis of hypoxic tumor mass [6]. So far, anti-angiogenic treatments are able to strengthen the viability of radiotherapy in lung cancer [1,7].

Angiogenesis plays crucial roles not only in tumor development but also in various physiological and pathological processes, including embryo development, normal growth, wound healing, and tissue repair [8]. Previous reports indicate that the response of endothelial cells to radiation determines the effects of tumor regression [5,9]. The response of endothelial cells is significantly influenced by stromal conditions such as macrophages and their secreted factors [5,6,10]. The effect of the endothelial response in tumor radiotherapy is not well understood, and understanding the detailed mechanism that enhances tumor angiogenesis is an important subject to improve the response.

Macrophages play an important role in inflammatory responses and tumor angiogenesis as they process tumor environmental cues, and directly secrete or stimulate other cell types to secrete mediators, including chemokines, cytokines and growth factors. Since macrophages are extremely plastic, they can be differently activated by various stimuli that together stimulate endothelial cell proliferation, degrade the extracellular matrix (ECM), and recruit leukocytes to further enhance angiogenesis [11].

Tumor associated macrophages (TAM) are commonly found in solid tumors and originate from circulating monocytes [12,13]. TAM are recruited to tumors by cytokine gradients produced by tumor cells as well as the tumor stroma [13] and promote tumor growth, often by producing a pro-angiogenic environment, and is correlated with poor clinical prognosis [8,14,15]. Additionally, an infusion of myeloid progenitor cells improves the tumor regrowth after local radiation [16]. While the generalized process of TAM recruitment has been identified, many unanswered questions and challenges remain.

Nerve injury-induced protein 1 (Ninjurin1 or Ninj1), is one of membrane proteins found on cell surface, that enhance the cell to cell adhesion [17,18,19]. Ninj1 was found to be up-regulated in blood-derived myeloid cells and blood-brain barrier endothelial cells during experimental allergic encephalomyelitis and in active multiple sclerosis lesions [20]. Other studies have also confirmed the functional role of Ninj1 in monocytes as well as in the adhesion regulators to endothelial cells [21]. It has been demonstrated that Ninj1 was induced by radiation in keratinocytes and dermal fibroblasts [22], and p53 contributes to the radiation-mediated Ninj1 up-regulation [23]. Considering the importance of p53 in radiation therapy, these findings suggest that Ninj1 might be involved in the radiation response of tumors. 

In this study, we report that radiation induces the Ninj1 expression in endothelial cells, and enhances adhesion/penetration of monocyte into endothelial cell adhesion. We further show that Ninj1 induction in endothelial cells due to radiation treatment is p53 dependent. These observations provide novel insights into the mechanisms involved in the pro-angiogenic effect of radiotherapy and unravel novel therapeutic perspectives for the improvement of current radiotherapy protocols.

## 2. Materials and Methods

### 2.1. Antibodies and Cell Cultures

Primary antibodies were procured as follows: Ninj1 (R&D Systems INC. Minneapolis, MN, USA), p53 and GAPDH (Santa Cruz Biotechnology, Santa Cruz, CA, USA), p21 and tubulin (Cell Signaling Technology, Danvers, MA, USA), and CD31 (Dianova, Hamburg, Germany). Human umbilical vein endothelial cells (HUVECs; Lonza, Walkersville, MD, USA) were maintained in a gelatin-coated dish with endothelial cell basal medium (Lonza) containing endothelial cell growth supplement. The human EA.hy926 hybrid cells and human monocytic cell line THP-1 cells were obtained from the American Type Culture Collection (Manassas, VA, USA), and I-HUVEC (immortalized HUVEC) was obtained from Applied Biological Materials (Richmond, BC, Canada). The cells were cultured in Dulbecco’s modified Eagle’s medium (DMEM) supplemented with 10% fetal bovine serum (FBS) and penicillin/streptomycin (GIBCO Invitrogen, Grand Island, NY, USA). The cells were irradiated with a dose of 1, 2, or 5 Gy using Novalis Tx™ (Varian, Palo Alto, CA, USA) equipped with a 2.5-mm width MLC. Irradiation was delivered using a 6-MV photon beam at a dose rate of 600 monitor unit/min (Mus/min), and the average required treatment time was 32, 64, and 160 s respectively, for 1, 2, and 5 Gy.

### 2.2. Western Blot Analysis

The preparation of whole-cell lysates, protein quantification, gel electrophoresis, and Western blotting were performed as described previously [19,24]. Briefly, proteins were extracted in lysis solutions with 20 mM Tris (pH 7.6), 1 mM EDTA, 140 mM NaCl, 1% NP-40, 1% aprotinin, 1 mM PMSF (phenylmethylsulfonyl fluoride), and 1 mM sodium vanadate. Protein concentrations were measured using a BCA protein assay (Pierce Biotechnology, Rockford, IL, USA). Equal amounts of cell lysate protein from each treatment group were resolved using SDS-PAGE and immunoblotted with primary antibodies. Antibody binding to the membrane was detected using enhanced chemiluminescence. Western blotting detection reagents were obtained from Absignal (Abclone, Seoul, Korea).

### 2.3. Animal Experiments

All animal procedures were conducted in accordance with a protocol approved by the Institutional Animal Care and Usage Committee at Gachon University in Incheon, Korea (LCDI-2015-0062). To determine the effect of radiation on Ninj1 expression in the tumor we used an in vivo mouse model as described previously [25]. A549 cells (5 × 10^6^ cells/100 uL PBS) were subcutaneously injected into the flank region of 6 weeks old female NOD/SCID mice. Eight mice were used and classified into 2 groups. Tumor volume was measured by using calipers and was calculated according to the formula (length × width^2^)/2. Three weeks after cell injection when the tumor volume reached about 300 mm^3^, tumors were treated with 5 Gy radiation. Next day mice were sacrificed, and tumor tissues were isolated, fixed in formaldehyde, and used for immunofluorescence analysis.

### 2.4. siRNA Transfection

Endothelial cells were trypsinized and harvested from a monolayer, and cells were suspended in an antibiotic-free medium. Complexes of siRNA and Lipofectamine RNAiMAX (Invitrogen, Carlsbad, CA, USA) were prepared as follows: 5 μL of 20 μM siRNA was diluted in 500 μL DMEM medium to which 5 μL of Lipofectamine RNAiMAX was added. After incubating this mixture for 20 min, the complexes were added to each well. Cells were incubated in a humidified CO_2_ incubator for 48 h. siRNA targeted against human p53 (5’-GAC UCC AGU GGU AAU CUA C-3’), and a scrambled non-targeting siRNA were synthesized by Bioneer (Daejeon, Korea). The siRNA-transfected cells were harvested for Western blot and RNA analysis.

### 2.5. Flow Cytometry Analysis

In order to evaluate the expression levels of Ninj1 by radiation, flow cytometry analysis was performed. Irradiated cells were suspended in PBS containing 0.05% Tween 20 and incubated for 30 min on ice in order to permeabilize the cell membrane. After washing with PBS, cells were incubated with anti-Ninj1 in PBS for 30 min on ice. After washing with PBS, cells were incubated with anti-Ninj1 antibody in PBS for 30 min on ice. Then the cells were stained with FITC-conjugated rabbit anti-goat IgG. Cells were subjected to flow cytometry analysis on a FACS Calibur cytometer (BD Biosciences, CA, USA). A total of 10,000 events were acquired and analyzed.

### 2.6. Immunofluorescence Analysis

To evaluate whether radiation mediates Ninj1 expression, sections were stained using anti-Ninj1 and anti-CD31 antibodies. Tumor tissue sections were deparaffinized and incubated with anti-Ninj1 and anti-CD31 antibodies, overnight at 4 °C. For antibody detection, we used Alexa fluor-488-conjugated goat anti-rabbit IgG (Life Technologies, Carlsbad, CA, USA) and TRITC-conjugated donkey anti-goat IgG (Santa Cruz Biotechnology, Inc) for 2 h. The slides were mounted with Vecta shield mounting medium and viewed under a confocal microscope (Nikon Eclipse Ti, Nikon NY, USA).

### 2.7. Adenovirus and Stable Cell Lines

Cells were seeded in 6-well tissue culture plates at a density of 3 × 10^5^ cells/well and were grown to 70–80% confluence before use. Complexes of adenovirus Ad-Ninj1 expressing human Ninj1 (Malvern, PA) and 20 mM final concentration of gadolinum (Gd3+) were made using gentle pipette tip aspiration, followed by incubation for 20 min at room temperature. Cells were washed with PBS, and the virus–Gd3+complexes were added to 2 mL serum-free medium. The amount of adenovirus was fixed at 10 and 50 MOI. After 4 h incubation in a CO_2_ incubator at 37 °C, cells were washed with PBS to remove the complexes, and were added to 2 mL fresh medium supplemented with 10% serum. Cells were then incubated for an additional 24 h. Adenovirus encoded shRNAs for human Ninj1 constructed in the pGIPZ vector were obtained from Thermo Fisher Scientific (Huntsville, AL, USA). Virus particles were generated in HEK293T cells transfected with the shRNA constructs and lentiviral accessory plasmids. THP-1 cells were infected with the virus by centrifugation at 2000 rpm for 2 h, and media change was given 6 h later. Since pGIPZ vector contains GFP gene, the cells showing GFP expression were sorted by FACS and amplified.

### 2.8. RT-PCR Analysis

To measure gene expression changes at the RNA level, reverse-transcription polymerase chain reaction (RT-PCR) was performed. Total RNA was isolated from cells using TRIzol reagent (Invitrogen, Carlsbad, CA). RT-PCR was carried out with gene-specific primers for Ninj1 (forward, 5’-CAA GTA CGA CCT TAA CAA CCC G-3’; reverse, 5’-AGG CCG TGA TGA AGA TGT TG-3’). Primers amplifying a region of GAPDH (forward, 5’-TTT GGT CGT ATT GGG CGC CTG-3’; reverse, 5’-CCA TGA CGA ACA TGG GGG CAT-3’) were used as internal controls. Reverse transcription PCR (RT-PCR) was performed in a T-100TM thermal cycler (Bio-Rad, Hercules, CA, USA) using AccuPower PCR Premix (Bioneer), according to the manufacturer’s protocol.

### 2.9. Zymography

The proteolytic activities of MMP-2, MMP-9, and uPA in CM (conditioned medium) were analyzed by substrate-gel electrophoresis using SDS-PAGE gels containing 0.2% (m/v) gelatin or 0.12% (m/v) fibrinogen and plasminogen (0.01 NIH unit/mL). I-HUVEC and THP-1 (SC or shNinj1) cell lines were co-cultured and the CM was concentrated using an Amicon Ultra-4 centrifugal device (Millipore, Bedford, MA, USA), and loaded onto gels. After electrophoresis, gels were washed with 2.5% Triton X-100 and incubated overnight in zymogram incubation buffer (50 mM Tris-HCl, 0.15 M NaCl, 10 mM CaCl2, and 0.02% NaN3) at 37 °C. Clear bands indicative of enzymatic activity were visualized by staining gels with 0.1% Coomassie blue staining solution.

### 2.10. Cell Binding Assay

Endothelial cells were irradiated or transfected with siRNA or infected with adenovirus. Two days later, GFP expressing THP-1 cells were washed twice with PBS and added to layer of endothelial cells. After incubation for 6 h at 37 °C, the wells were gently washed twice with warm medium to remove non-adherent cells. The wash medium was discarded, and digital images of the THP-1 (GFP) cells bound to the endothelial cells were acquired using an Incucyte (Essen Bioscience, Ann Arbor, MI, USA). The imaging system was performed in a CO_2_ incubator. The ratio of adherent THP-1 cells to endothelial cells was calculated as the ratio of GFP-positive cells to total cells.

### 2.11. In Vitro Transmigration Assay

Transwell inserts of 8 μm pore size (Corning Incorporated, Corning, NY, USA) were inverted, and the filter was plated in 50 µL of 5 μg/mL growth factor reduced matrigel (BD) in DMEM media for 1 h at room temperature. Adenovirus Ad-Ninj1 infected I-HUVECs (1 × 10^5^ cells) were plated on the matrigel-coated filters on the inverted transwells in 50 μL DMEM for 4 h in a 37 °C CO_2_ incubator. Transwells were then flipped right side up into a 24 well plate, and co-cultured with THP-1 (GFP) cells for 10 h. To assess the extent of THP-1 cell on the transendothelial migration, transwells were washed twice with PBS, fixed in 4% paraformaldehyde/PBS for 20 min, and washed. After mounting the slides with VectaShield Mounting Medium containing DAPI as the nuclear stain (Vector Laboratories, Burlingame, CA), the slides were sealed with nail polish and stored at −20 °C, until observation using a confocal microscope (Nikon Instruments Inc., Melville, NY, USA). Only those THP-1 (GFP) cells that breached the endothelium were scored as a positive transendothelial migration event. Approximately 6 fields of view were acquired for each transwell.

### 2.12. Statistical Analysis

Data are shown as means ± standard deviations. For statistical analysis of comparative data, a two tailed Student’s t-test was utilized using Microsoft Excel software. Values of * *p* < 0.05; ** *p* < 0.01; *** *p* < 0.005 were considered as significant difference and indicated by asterisks in the figures.

## 3. Results

### 3.1. Surface Expression of Ninj1 Following Treatment with Radiation in Endothelial Cell Lines

Radiation treatment is known to enhance the p53 dependent Ninj1 expression [22,23]. To determine whether radiation treatment up-regulates Ninj1 expression in endothelial cell lines such as HUVEC, EA.hy926, and I-HUVEC, we first examined the expression of Ninj1 in radiation-treated endothelial cells. We found that expression of the Ninj1 protein and mRNA were increased in the three endothelial cell lines (Figure 1A). Next, we sought to determine the cellular location of Ninj1 in the endothelial cell lines following radiation treatment. EA.hy926 and I-HUVECs were stained for Ninj1 and analyzed by flow cytometry following different doses of radiation treatment (1, 2, and 5 Gy). We found that the induction of surface Ninj1 in irradiated endothelial cell lines corresponded to the radiation dose (Figure 1B). To investigate whether radiation-treated endothelial cells increased the expression of Ninj1 in vivo, A549 human lung adenocarcinoma cells were grafted in nude mice. When the tumor volume reached 300 mm^3^, tumors were treated with 5 Gy radiation. The Ninj1 expression was assessed on the basis of immunofluorescent staining with antibodies against CD31 and Ninj1 on the surface of endothelial cells. Immunofluorescent staining of CD31/Ninj1 double positive cells, which suggests the presence of endothelial cells, was abundantly found in the radiation treated tumors (Figure 1C). These results provided evidence for the irradiation-enhanced expression of Ninj1 on the surface of endothelial cells.

### 3.2. p53 Enhanced Ninj1 Expression in Endothelial Cell Lines

Radiation temporarily induces DNA damage resulting in activation of p53 in endothelial cells. After irradiation, Ninj1 is transcriptionally activated by p53, which is induced by DNA damage in human lung cancer cell lines [23]. However, the response of p53 to radiation varies in different cell lineages. To determine the function of p53 after radiation treatment of endothelial cells, HUVEC, EA.hy926, and I-HUVEC endothelial cell lines were irradiated with indicated dose of radiation. The p53 protein has a short half-life; however, it was increased through the radiation-mediated DNA damage response (Figure 2A). Simultaneously, the protein level of Ninj1 was also markedly increased by radiation. These results suggest that the Ninj1 expression depends on the functional status of p53 in endothelial cells. To investigate this possibility in further detail, we transfected scramble or p53 siRNA to EA.hy926 and I-HUVEC endothelial cell lines and compared the changes between the transfected cells after irradiation. RT-PCR analysis revealed that Ninj1 was decreased in p53 knock-down as compared with scramble siRNA transfected endothelial cell lines exposed to the same dose of radiation (Figure 2B). Western blotting analysis revealed that the level of the Ninj1 protein was suppressed in endothelial cell lines with p53 siRNA transfection (Figure 2C). Together, these findings indicate that p53 activity in endothelial cells regulates the Ninj1 expression. These results support that over-expression of Ninj1 in irradiated endothelial cells easily captures the circulating monocytes, which may be induced by an altered microenvironment after radiotherapy.

### 3.3. Over-Expressed Ninj1 in Endothelial Cells Enhanced Monocyte Attachment

Upon activation, the attachment of monocytes to endothelial cells is a distinct endothelial cell function, which is a critical event in radiation-induced influx of monocytes into tumors [13]. Hence, we explored if radiation-induced Ninj1 can influence monocyte attachment to endothelial cells. We first quantified the number of GFP labeled THP-1 cells that adhered to the endothelial cell lines. The EA.hy926 and I-HUVEC endothelial cell lines were grown to sub-confluence, irradiated with indicated dose of radiation, co-cultured with GFP labeled THP-1 cells for 4 h, after which the unattached cells were gently washed out. The adherent THP-1 cells were examined by IncuCyte live cell analysis system, and digital images were captured at ×200 magnification. We found markedly enhanced adhesion between monocytes and endothelial cell lines following the radiation treatment (Figure 3A,B). As previously shown, p53 regulates Ninj1 expression after irradiation in endothelial cells. Hence, we explored if p53 has a role in mediating the radiation treatment enhanced binding of the monocytes and endothelial cells. The p53 siRNA transfection to I-HUVEC significantly attenuated radiation-mediated THP-1 and I-HUVEC binding (Figure 3C). To further understand the role of Ninj1 in endothelial cell and monocytes attachments, we sought to determine if ectopic expression of Ninj1 induces endothelial cell and monocyte attachments. EA.hy926 and I-HUVEC endothelial cell lines grown to a sub-confluent monolayer were transfected with Ninj1 encoded adenovirus (AdNin) for 24 h, followed by addition of GFP labeled THP-1 cells for 4 h, after which the unattached cells were washed out. The adherent THP-1 cells were examined by IncuCyte live cell analysis system. We found that ectopic expression of Ninj1 in endothelial cell lines induced the attachment of monocytes (Figure 3D,E). Of note, radiation-treated endothelial cells significantly enhanced the Ninj1-induced binding between the monocytes and endothelial cell lines in a dose-dependent manner. In our results, Ninj1 is considered as the one of main adhesion molecules that mediates the radiation-induced adhesion of monocytes to endothelial cells. Therefore, we undertook to determine whether Ninj1 to Ninj1 homophilic binding directly affects the adhesion between monocytes and endothelial cells. The THP-1 human monocyte cell lines were transfected with Ninj1 shRNA, and we established a stable transfected cell line (Figure 4A). The monocyte adhesion assay was further conducted in the co-culture system of endothelial cell lines and stable Ninj1 shRNA THP-1 cell lines. Sub-confluent EA.hy926 and I-HUVEC endothelial cell lines were irradiated and incubated with stable control or Ninj1 shRNA THP-1 cell lines for 4 h. The stable Ninj1 shRNA cell lines showed lower endothelial cell binding activity compared with stable control shRNA cell lines (Figure 4B,C). This inhibitory activity of Ninj1 knock-down in THP-1 to adhesion of irradiated endothelial cells may result from the homophilic binding activity of surface Ninj1 proteins. Thus, we suggest that monocyte adhesion to endothelial cells was augmented by radiation treatment, which was increased by Ninj1. This study attempted to clarify that Ninj1 expression was radiation dependent in endothelial cells. Our results irrevocably indicate that in endothelial cells exposed to radiation, the monocyte binding was increased by Ninj1 to Ninj1 homophilic binding activity.

### 3.4. Over-Expressed Ninj1 in Endothelial Cells Increases Monocyte Transmigration and Activation

Several lines of evidence suggest the involvement of Ninj1 in the endothelial cell and monocyte adhesion during tumor infiltration. To better define the role of Ninj1 in transmigration, transwell chambers were coated with sub-confluent I-HUVEC endothelial cell lines transfected with AdNin. Transmigration of THP-1 cell lines through Ninj1 over-expressed in I-HUVEC was higher than observed in control adenovirus transfected I-HUVEC (Figure 5A), whereas stable Ninj1 shRNA THP-1 cell line showed a marked decrease of bound cells under similar coating conditions (Figure 5B). Our results suggest that the mechanism by which Ninj1 in endothelial cells contributes to monocyte extravasation involves intercellular adhesion, mediated after radiation treatment. This strong adhesion is probably because of the homophilic binding of Ninj1 in endothelial cells and monocyte. Next, the adhesion of monocytes to endothelial cells was studied in monocytes activated with MMP secretions. This expression was similarly inhibited by Ninj1 knock-down in the monocyte cell line (Figure 5C). Our data supports the involvement of Ninjuirin1 in monocyte infiltration through endothelial cells, a process that is stimulated by radiation treatment.

## 4. Discussion

Previous studies have suggested that radiation enhances the hematopoietic cell adhesion, migration and invasion, simply by affecting the endothelial cell surface adhesion molecules, including Ninj1 [5,6,16,23]. The present work aims to understand the effect of radiation on endothelial vascularization and monocytes, as they are the important components of the tumor microenvironment, and are highly recruited into tumors during radiation therapy [26]. The endothelial cell apoptosis has been suggested to mediate tumor responses after radiation [27]. Many groups reported radiation induces p53 in endothelial cells from various sources in vivo and in vitro [28,29]. Ninj1 has been reported a target of p53 and found to suppress of p53 translation [23,30], suggesting that Ninj1 functions to protect cancer cells against radiotherapy, because endothelial cell mediated myeloid cell recruitment a crucial role in initiating radiation induced tumor tissue remodeling [31]. Moreover, understanding whether Ninj1 plays a pivotal role in irradiated endothelial cells will provide critical information to design clinical trials that combine radiation therapy with inhibitors of the DNA damage response pathway. Here, we investigated the functional role of Ninj1 in enhancing endothelial cell and monocyte adhesion and transmigration, by simple experiments involving radiation-mediated up-regulation of Ninj1 in endothelial cells.

Our results indicate the involvement of Ninj1 in monocyte extravasation by binding to endothelial cells, a process that is stimulated by radiation treatment (Figure 3). Indeed, p53 was found to be a selective and potent regulator of Ninjuirin1 expression in endothelial cells, which is a promising tool for angiogenesis as well as a therapeutic target. Surface Ninj1 was increased by radiation treatment, whereas p53 knock-down blocked the Ninj1 expression in endothelial cells (Figure 2). 

During the radiation therapy, the phenomenon of resistance has been observed [2,5]. Increased tumor infiltrated monocytes have been described in radiotherapy. Monocytes are recruited as a first response to radiation- induced damage in the tumor microenvironment [31]. Several studies have reported that macrophage accelerate the tumor remodeling via the tumor vasculature and development of tumor regeneration. The findings of this study support a model for radiation-induced Ninj1 and tumor remodeling. Ninj1 from HUVEC, EA.hy926 and I-HUVEC cells induced recruitment of monocytes, resulting in vasculature after radiation. We initially expected that the induction of Ninj1 by the radiation would improve the tumor infiltration of macrophage. Also, the actual results indicated that Ninj1 inhibition by shRNA significantly decreased both endothelial cell binding and MMPs secretion.

Considering the above results, radiation treatment appears to enhance the production of Ninj1 via increased transcription of p53 after irradiation. The shRNA-mediated knockdown of Ninj1 was also used to specifically examine its role in monocyte adhesion and migration. To the best of our knowledge, this is the first report to clearly demonstrate that radiotherapy stimulates the monocyte and endothelial cell interaction through Ninj1 homophilic binding. Furthermore, an interesting observation of this study is the ability of Ninj1 deficiency in monocytes to inhibit endothelial dependent monocyte activation (Figure 5). This observation may contribute to understand the possible pathogenic role of the up-regulated levels of Ninj1 in several pathologies, including development, cancer and colitis [24,32,33]. The impact of this adhesion activity on endothelial cells is probably related to the monocyte-mediated inflammatory action reported for the pathogenic role in inflammatory diseases such as sepsis [34], and for the regulatory role of Ninj1 in angiogenesis [35], suggesting that this protein is a promising therapeutic target in cancer. 

Usually, radiation treatment causes tissue injury, followed by a pathological repair process in the tissue, including angiogenesis, ECM deposition and tissue remodeling [36]. Studies have reported that MMPs are involved in the metastasis and angiogenesis of tumors as well as numerous other pathological processes [37,38]. In the present study, we examined the effect of the homophilic binding of Ninj1 on the production of MMP-2 and MMP-9 in monocytes. The tumor microenvironment has influence on the tumor recurrence. In the present study, we examined the effect of the homophilic binding of Ninj1 on the production of MMP-2 and MMP-9 in monocytes. Overexpression or aberrant MMPs expression is positively correlated with tumor progression and recurrence [38]. The two MMP-2 and MMP-9 have been associated with the malignant phenotype of tumor cells because of their unique ability to degrade type IV collagen, which is a major component of the basement membrane [39]. In addition, activated MMP-2 and MMP-9 is associated VEGF bioavailability in cancer [40]. Eventually, the number of recruited monocytes in the tumor through endothelial cells has increased via Ninj1 binding in our results. The monocyte infiltration then promotes vasculature and tumor cell proliferation that characterizes radiotherapy resistance [16].

In this study, we observed that the irradiated endothelial cells dramatically enhanced the production of MMP-2 and MMP-9 in co-cultured monocytes. This result supports the notion that ionizing radiation enhances the production of MMP-2 and MMP-9 through the over-expression of the Ninj1 protein (Figure 5). In contrast, the levels of MMP-2 and MMP-9 in Ninj1 knock-down monocytes did not change on exposure to irradiated endothelial cells. Homophilic binding of Ninj1 is therefore likely to stimulate the MMP-2 and MMP-9 expression in monocytes

In conclusion, we demonstrated in this study that ionizing radiation enhances the expression of Ninj1 in endothelial cells. In addition, our data indicates that p53 is a potent regulator of Ninj1 expression in endothelial cells, suggesting that p53 inhibitors would be beneficial for the treatment of radiation therapy recurrence. Our work also adds valuable data on characterization of tumor recruiting of monocytes subjected to radiation treatment, as used during cancer treatment. We have characterized important aspects like endothelium adhesion, extravasation, and tumor remodeling, which promote tumor recurrence. Based on these findings, we found that radiation treatment increases the monocyte infiltration and promotes angiogenesis, thereby supporting radiation resistance. Taken together, the results of the present investigation support the hypothesis that Ninj1 produced by endothelial cells may be deeply involved in tumor radiotherapy resistance. However, more detailed studies are required in order to further evaluate the mechanisms underlying the observed effects.

## 5. Conclusions

The active role played by Ninj1 in irradiated tumors is compatible with: (1) The over-expression of Ninj1 in endothelial cells, characterized by facilitating circulating monocytes to directly enter tumor; (2) the increased expression of Ninj1 in endothelial cell facilitating tumor re-modeling after radiation treatment through monocyte recruitment.

## Figures and Tables

**Figure 1 cells-09-01086-f001:**
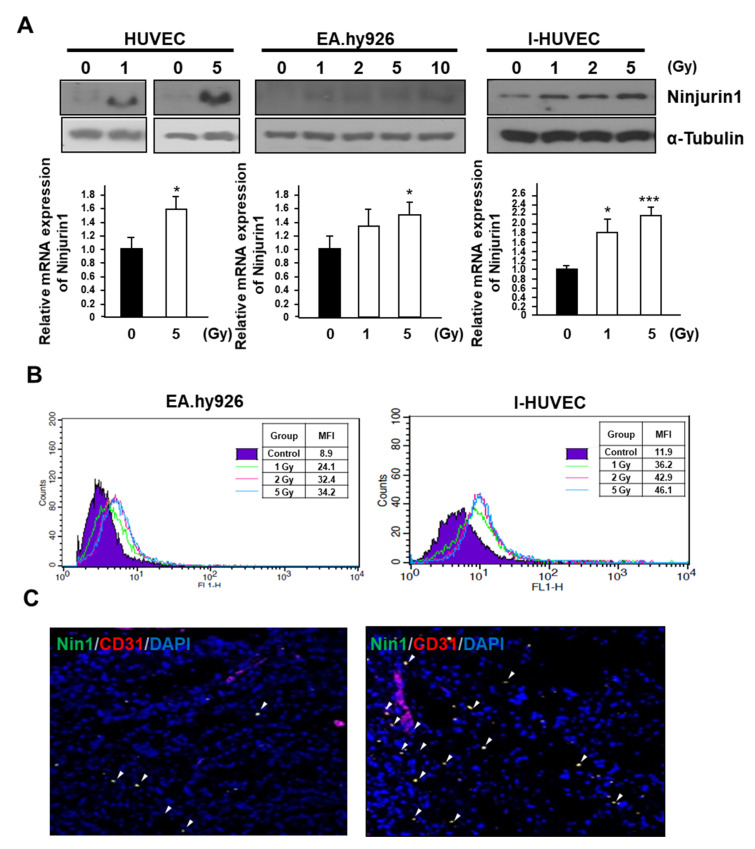
Radiation enhances Ninj1 expression in endothelial cell lines. (**A**) Following exposure to radiation, human endothelial cell lines were cultured for an additional 24 h and analyzed for Ninj1 protein and RNA expression. (**B**) FACS histogram of surface Ninj1 expression in human endothelial cell lines, 24 h after indicated dose of radiation treatment. (**C**) Representative image of immunofluorescence double staining for endothelial cell (CD31; Red) and Ninj1 (Green) in frozen sections of A549 human lung adenocarcinoma xenograft tumor. Arrow heads indicated co-localization of CD31 and Ninj1. Significant difference from control samples: * *p* < 0.05; ****p* < 0.001.

**Figure 2 cells-09-01086-f002:**
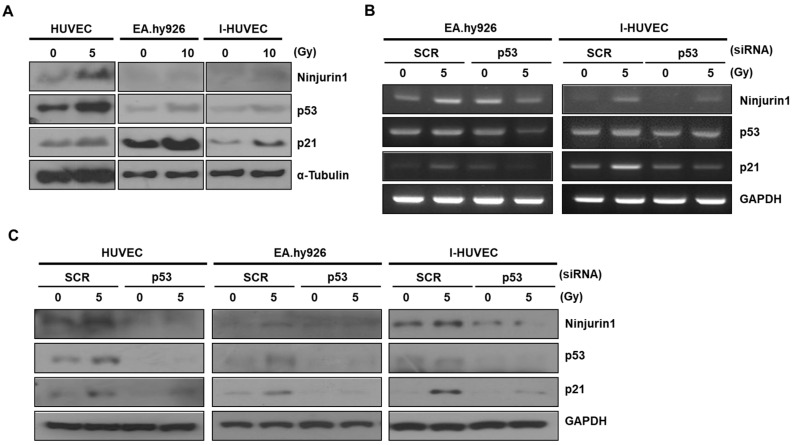
Radiation-mediated p53 transactivation enhances Ninj1 expression. (**A**) The irradiated human endothelial cell lines were subjected to western blotting and evaluation of p53 and Ninj1 protein expression. (**B**) Ninj1 expression was analyzed with semi-quantitative RT-PCR. Cells were treated with radiation after transfection with control or p53 siRNA to determine effective p53 knock-down after radiation treatment. (**C**) Western blotting confirms the suppression of Ninj1 expression with p53 siRNA transfection.

**Figure 3 cells-09-01086-f003:**
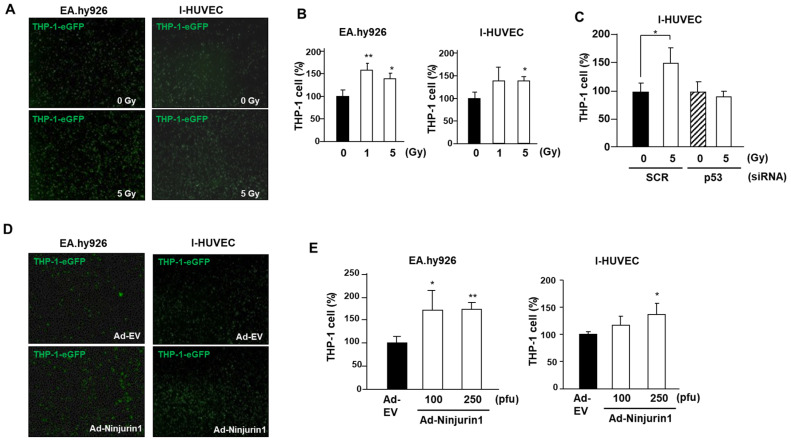
Radiation exposure enhanced binding of endothelial cells and monocytes by Ninj1. (**A**) Endothelial cell lines were treated with 5 Gy radiation and incubated for 24 h. GFP labeled THP-1 monocytes were added to a monolayer of irradiated EA.hy926 for 4 h, and then washed out for unattached cells. Representative experiment is illustrated out of three performed. (**B**) Quantification of GFP labeled THP-1 cells bound to the endothelial monolayers was carried out by measuring the surface area of the cells using ImageJ software. Percentages of the cell signal with respect to the total area are indicated. (**C**) I-HUVECs were transfected with control or p53 siRNA for 24 h, after which they were exposed to 5 Gy radiation. Transfected I-HUVECs were incubated with GFP labeled THP-1 cells for 4 h. We observed for attachment of THP-1 cells to I-HUVECs after washing. Quantitative analysis showing the attachment of THP-1 cells to I-HUVECs monolayer in the condition of knock-down of p53. (**D**) To examine THP-1 cell adhesion to ectopic Ninj1 over-expressed endothelial cell lines, the endothelial cell lines were transfected with indicated dose of Ninj1 encoded adenovirus and incubated for 24 h and co-cultured with GFP labeled THP-1 cell for 4 h. Non-adherent cells were then washed out, and adherent cells were visualized using IncuCyte. (**E**) The histogram on the left represents quantitative results of image obtained from ImageJ software. Data are presented as the mean ± s.d. Significant difference from control samples: * *p* < 0.05; ** *p* < 0.01, respectively by Student’s t-test analysis.

**Figure 4 cells-09-01086-f004:**
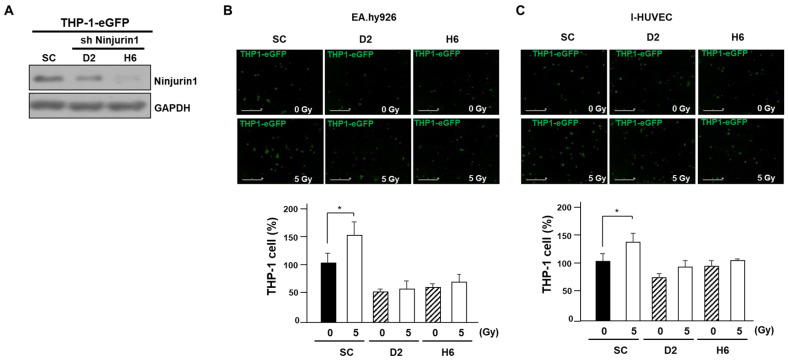
Ninj1 molecules required homophilic interaction between endothelial cells and monocytes. (**A**) Ninj1 protein expression was detected in stable Ninj1 shRNA transfected cells. Ninj1 was knockdown by shRNA. (**B**,**C**) Knockdown of Ninj1 in monocytes by shRNA inhibited adhesion to irradiated endothelial cell lines including EA.hy926 cell (**B**) and I-HUVEC (**C**). Confluent endothelial cell lines were treated with selected dose of radiation and were incubated with stable Ninj1 knock-down THP-1 cell lines for 4 h. Non-adherent cells were then washed away, and attached cells were observed with IncuCyte. Quantitative analysis revealed the attachment of Ninj1 knock-down THP-1 cells to irradiated endothelial cell lines monolayer. Data are presented as the mean ± s.d. Significant difference from control samples: * *p* < 0.05, respectively by Student’s t-test analysis.

**Figure 5 cells-09-01086-f005:**
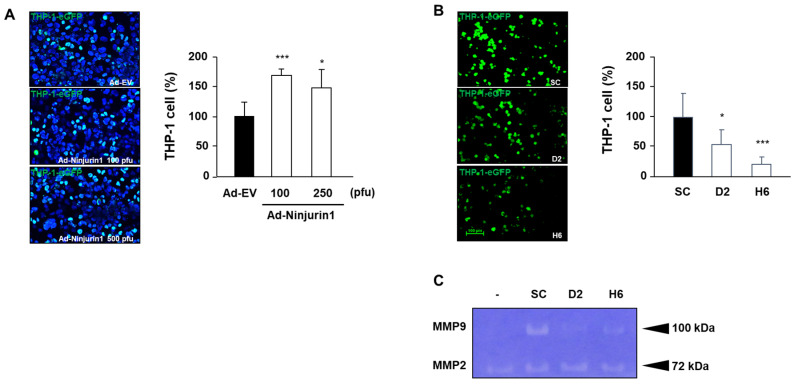
Effect of Ninj1 on the transmigration and activation of monocytes. (**A**) Confluent I-human umbilical vein endothelial cell (HUVEC) lines were transfected with Ninj1 encoded adenovirus and cultured for 24 h on transwell chamber coated with 0.1% gelatin on its lower surface. Subsequently, GFP labeled THP-1 cells were added to the wells and incubated for 12 h. THP-1 cells migrating to the lower compartment of transwell chamber were microscopically observed. Quantitative analysis show the migrated THP-1 cells in the lower bottom of transwell chamber. (**B**) Confluent I-HUVEC endothelial cell lines were cultured in transwell chamber coated with 0.1% gelatin on its lower surface. Subsequently, Ninj1 knock-down THP-1 cells were added to the well and incubated for 12 h. Quantitative analysis show the migrated THP-1 cells in the lower bottom of transwell chamber. (**C**) Gelatin zymography of the conditioned medium obtained from co-cultured endothelial cell and Ninj1 knock-down monocyte. The gelatinolytic band of 100 and 72 kDa corresponds to pro-MMP-2 and pro-MMP-9, respectively. Significant difference from control samples: * *p* < 0.05; ****p* < 0.001.

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
