# Peer review of "Radiation Potentiates Monocyte Infiltration into Tumors by Ninjurin1 Expression in Endothelial Cells"

_cells, 2020, doi:10.3390/cells9051086_

Round 1
Reviewer 1 Report
Kang JH et al. demonstrated that radiation induces Ninj1 expression in tumor-associated endothelial cells, and thus promotes macrophage adhesion and infiltration to tumor tissue. The experiments seem to be well performed and support the conclusions. There remain a few issues for this reviewer.
Figure 1A: Please indicate Y axis units.
Figure 1A: Please check Y axis unit of I-HUVEC data. Distances between 0 and 0.5 (i.e. 0.5 unit) and between 0.5 to 1.5 (i.e. 1.0 unit) seems to be the same.
Figure 1B: Provide quantification data. For example, mean fluorescence intensities (MFI) values can quantitatively represent the data.
Figure 1C: The images are not comprehensible in its current format. Increase intensities/contrasts or enlarge images for better comprehension.
Figure 2B: Why do the authors think that p53 bands appear in si-p53 reated cells?
Figures 3-5: THP-1 cells are not macrophage cell lines (as written multiple times in the manuscript), but human monocytes or rather acute monocytic leukemia cells. THP-1 are widely used as macrophages for in vitro experiments, but only with proper cytokine stimulants. This is the limitation of this work, and should be discussed in the discussion.
Reviewer 2 Report
The study demonstrates that ionizing radiation enhances the expression of Ninjurin1 in endothelial cells. The authors confirm that p53 is a potent regulator of Ninj1 expression in endothelial cells, suggesting that p53 inhibitors would be beneficial for the treatment of tumor recurrence after radiation therapy.
The topic of the paper is interesting and the data seems reliable. However, I have some suggestions.
line 37-38, “dependent on damage to genomic DNA, which produces highly reactive free radicals” – do you mean that damage to DNA produces radicals or maybe ionizing radiation is responsible for highly reactive radicals?
line 97, could you explain how from the dose rate of “600 monitor unit/min” you calculated the absorbed dose at Gy. How long were the samples irradiated to get 1, 2 or 5 Gy?
line 104, I suggest using simply SDS-PAGE instead “sodium dodecyl sulfate-polyacrylamide gel electrophoresis”;
line 112, please, properly cite {Kang, 2019 #224};
In Section 2.3 please provide the exact protocol of experiments on animals. How many animals did you examine in every group?
line 130, was anti-Ninj1 conjugated with fluorescent molecule for cytometric measurements (FACS)?
line 207, I suggest using radiation dose instead radiation strength.
Fig. 2B, is it really seen for I-HUVEC in Fig. 2B that Ninj1 was decreased in p53 knock-down cells? I see that only for EA.hy926.
Fig. 4, in the text there are only 4a and 4c, and in the Figure caption there are 4a and 4b, however in Fig. 4 there are a, b, c. Please correct.
I also suggest considering the results in the paper: Ninjurin 1 has two opposing functions in tumorigenesis in a p53-dependent manner, Hee Jung Yang, Jin Zhang, Wensheng Yan, Seong-Jun Cho, Christopher Lucchesi, Mingyi Chen, Eric C. Huang, Ariane Scoumanne, Weici Zhang, and Xinbin Chen. PNAS, 2017, 114 (43), 11500-11505; doi:10.1073/pnas.1711814114, in the Discussion section. I think the studies are compatible and could help to prove the Authors’ conclusions.
